# Renewable Energy Communities as Modes of Collective Prosumership: A Multi-Disciplinary Assessment, Part I—Methodology

Shubhra Chaudhry [1,†], Arne Surmann [2,*,†] , Matthias Kühnbach [2] , Frank Pierie [1]

1    Institute of Engineering, Hanze University of Applied Sciences, 9747 AS Groningen, The Netherlands
2    Department of Smart Grids, Fraunhofer Institute for Solar Energy Systems ISE, 79110 Freiburg, Germany
*    Correspondence: arne.surmann@ise.fraunhofer.de; Tel.: +49-761-4588-2225
†    These authors contributed equally to this work.

**Abstract:** Citizens are set to play an active role in the energy transition by transforming from 'passive' consumers to 'active' prosumers. Renewable Energy Communities (RECs) are envisioned as modes of collective prosumership by citizens under the Renewable Energy Directive of 2018 (RED II). A holistic understanding of RECs is essential to identify the benefits and challenges of collective prosumership. RECs have been the topic of several modelling studies, but a single model that simulates RECs from an integrated perspective—combining technical, economic and ecological analysis—is absent. Wide variability in the indicators discourages comparison of the results across studies. This article builds on the existing knowledge by proposing an integrated model to undertake a multi-disciplinary assessment of a potential REC. First, the proposed model analyses the technical possibilities of collective prosumership using energy flow analysis based on consumption and generation profiles. Second, the model evaluates the economic impacts of prosumership from two perspectives: from the consumers' perspective (in terms of the annual cost of energy consumption) and from an investor's perspective (in terms of the net present value of the investment). Thirdly, the model quantifies the annual greenhouse gas emissions of energy consumption (expressed in $CO_2$ equivalent) to evaluate the ecological impact of prosumership. Lastly, a set of key performance indicators (KPIs) are proposed that can be used to interpret and compare the results of simulations and are mapped to the actors in the REC in line with their objectives. The proposed approach offers a single, replicable model that can be used to simulate RECs in the different Member States of the European Union. The KPIs can be used to compare the impact of combinations of various prosumership activities within the same REC or to compare two different RECs on the benefits offered vis-a-vis the investments incurred. The KPIs also offer insights into the aligning and conflicting objectives of the stakeholders of the REC.

**Keywords:** renewable energy communities; prosumer; self-consumption; energy sharing; self-sufficiency

## 1. Introduction

The emergence of local energy systems in Europe is changing who owns, generates and distributes energy. Citizens are transitioning away from being passive consumers and taking an active role as prosumers and co-owners of distributed energy systems [1]. Prosumers are a new type of energy users—ones who consume, produce, store and share energy with other energy users [2]. Active participation in renewable energy (RE) and prosumership are cornerstones of a successful and democratised energy transition [3,4]. Individuals pursuing prosumership offer a host of benefits such as higher decentralised energy production, energy storage, demand response and demand reduction [2]. Community-owned energy projects with collectively prosumership further enhance the positive impact of individual prosumers through maximised local consumption of RE, efficient energy transfer, lower energy losses and aggregated potential for demand-side management [2]. Finally, by consuming higher shares of locally generated energy, citizens may also benefit from lower prices.

As part of the Clean Energy for All Europeans package [5], the European Union (EU) attempted to homogenise and harmonise the diversity of community energy projects across the Member States. The legal basis for collective energy prosumership was defined under the concept of Energy Communities (ECs) in two Directives: the Renewable Energy Communities (RECs) defined in the Renewable Energy Directive [6] and the Citizen Energy Communities (CECs) defined in the Internal Electricity Market Directive [7]. The transposition of these directives is in progress with varying degrees of completion across Member States [8].

### 1.1. State of Science

Renewable Energy Communities, and energy communities in general, are of relevance to multiple disciplines, as a result of which scientific studies have looked at them from one or more aspects, such as the technical, economic and ecological.

Technical studies on energy communities covered areas such as the feasibility of different technologies and reviews of technologies that enable the functioning of energy communities. Syed et al. [9] conducted a study to reduce the electricity consumption from the grid by using solar PV and battery storage installations in residential apartments in Australia. By using different setups of microgrids at three sites, the self-sufficiency of energy demand for the sites was shown to be greater than 60%. Gul et al. [10] propose a new technology involving a combined heat and power system to maximise the utilisation of RE produced in two communities in Italy. The proposed methodology aims to maximise the generation of RE, to balance the load with supply from local RE and grid energy and, lastly, to balance the heat demand from locally produced heat and boilers. The study then evaluates the economic and environmental impact of the proposed technology by evaluating the net present value ($NPV$), the levelised cost of energy (LCOE) and the reduction in $CO_2$ emissions. The proposed system met 79% of the energy demand with the lowest LCOE (0.013 EUR/kWh) and reduced 4129 tons of $CO_2$ per year. Menniti et al. [11] conducted a review of the technologies and services that enable the operations of RECs. Enabling technologies included those for measuring and monitoring energy flows and managing energy systems. The study shows that using the enabling technologies increases the self-consumption of energy, which can be further increased by using more technologies together. Furthermore, self-consumption is higher when end users act together in an REC compared with when they act as individual prosumers. Purely economic studies on energy communities were less common and focused on financial feasibility investments or quantifying the costs of prosumership. Petrichenko et al. [12] studied the profitability of collective versus individual prosumership in Latvia. The study found that acting as an energy community avoided 20% greater energy costs and cut the payback period in half compared with prosumers acting alone.

The technical and economic aspects of energy communities were found to be closely linked, evidenced by the volume of studies with a techno-economic lens. Moncecchi et al. [13] proposed a techno-economic model to optimise RE production and to identify the optimum investment decision for members of an Italian energy community based on different incentives. They conclude that incentives for energy sharing led to an optimally sized system, while incentives for energy production led to an oversized system. Moreover, systems based on only energy production had higher $NPV$ than those based on self-consumption and energy sharing. Viti et al. [14] undertook a techno-economic analysis of building clusters in northern Italy and compared the benefits of prosumership under two modes: individually versus collectively via energy communities. The study concluded that the benefits of collective prosumership outweigh individual prosumership, observed in higher self-consumption and self-sufficiency rates and higher savings in energy costs. Higher self-consumption, however, leads to lower $NPV$ due to the loss of revenue due to lower sales of electricity to the grid. Jasiński et al. [15] studied how rural energy cooperatives could be organised under the regulatory framework in Poland. Using five energy cooperatives as case studies, the authors optimised the production mix of the RE technologies and calculated the total energy consumption. All cases showed an increase in

self-consumption, while three out of five cooperatives had a reduction in energy consumed from the grid. The economic analysis found that four out of five energy cooperatives were profitable, with profitability ranging between 7.7% and 27.5%. Huang et al. [16] proposed a top-down control methodology to optimise the system operation of a cluster of nearly zero energy buildings. The study concluded that coordinating the collaboration between individual buildings within the group achieves a higher self-consumption by increasing the sharing of energy within the cluster, which leads to lower consumption from the grid and, ultimately, lower energy costs. Norbu et al. [17] proposed a techno-economic modelling methodology that combines RE with battery storage operating within the constraints of grid capacity and battery degradation. A comparison of individual investment in RE assets is made vis-a-vis collective community ownership assets. Using real data from a case study in the United Kingdom, the study finds that individually operated assets are less impacted by grid constraints, shown by lower curtailment of assets, whereas community-owned assets face higher curtailment, which results in higher energy bills for consumers. The study emphasises the need to study the impact of physical grid limits on the operation of community-owned RE assets. In another case study from the United Kingdom, Kastel et al. [18] studied the techno-economic viability of aggregating load and production profiles of consumers in six locations. Aggregation of load profiles led to higher shares of energy self-consumption, ranging between 31–72%. The pooling of energy assets also had a positive impact on the LCOE and enabled 4% of solar PV sites to achieve parity with grid prices. Bartolini et al. [19] focused on a district in the United States and studied the role of battery storage and energy conversion technologies in integrating high shares of variable RE. The study compared a battery-only scenario with a multi-energy system comprising energy conversion and storage and concluded that the multi-energy systems are the least-cost option for achieving the goal of locally consuming all RE energy. He et al. [20] investigated the techno-economics of two microgrid scenarios—standalone and grid-connected—for an RE-based residential community in Beijing, China. The most cost-effective configuration of the energy system was evaluated in the optimisation study. The RE-based microgrids could lead to a self-sufficiency of 90% of the demand in the community and consume 47–100% of the RE locally. The grid-connected energy system was found to have a net present cost reduction of 57% compared with consuming all the energy from the grid. The inclusion of battery storage was found to improve the cost-effectiveness of standalone and grid-connected microgrids.

The economic and ecological aspects of prosumership and energy communities have also been researched. Fleishacker et al. [21] researched the economic and ecological impacts of forming an energy community in Austria and concluded that forming energy communities could lead to a reduction in system costs and greenhouse gas (*GHG*) emissions, but these goals could not be achieved simultaneously and were inversely related, that is, minimum system costs were accompanied by maximum *GHG* emissions and vice versa. Dal Cin et al. [22] used the bi-objective methodology to achieve the two competing goals of minimum system costs and minimum *GHG* emissions for three configurations of an energy community. The study of an Italian case concluded that collectively organising as an energy community can lead to a 12% reduction in energy system cost and a 21% reduction in *GHG* emissions when compared with the case without any prosumership by passive consumers.

Ceglia et al. [23] undertook an analysis of an REC using a techno-ecological perspective. In the technical part of the analysis, the energy flow analysis of two buildings with solar PV and electric vehicles was carried out to compare the energetic advantages of forming an REC instead of acting as individual prosumers. By sharing energy as an REC, 79% of the PV is consumed locally. The ecological analysis evaluated the $CO_2$ emissions of the REC due to electricity consumed from the grid and undertook a comparison by using average grid emission factors and time-varying emission factors based on the actual production mix for the grid. Using time-dependent emission factors led to a reduction of 12% in the $CO_2$ emissions of the REC, highlighting the necessity of using variable factors when evaluating energy systems based on variable energy sources. Schram et al. [24] looked at

the impact of energy communities based on four different technologies in reducing the national *GHG* emissions across eight nations of the EU. The *GHG* reduction potentials of electric vehicles, photovoltaic, heat pumps and batteries were quantified across Austria, Belgium, France, Germany, Italy, the Netherlands, Portugal and Spain. Households acting as energy communities and sharing energy led to higher PV capacities across nations when compared with households acting as individual prosumers, which enhanced the *GHG* reduction potential of all technologies. All technologies under study reduced *GHG* emissions, and the reduction was higher in nations with fossil-fuel-based energy systems.

A wide variety of indicators that have been used to assess different aspects of RECs has been observed across the literature. The most common indicators used for technical aspects are self-consumption [11,14–16,18–20,23] and self-sufficiency [9,10,14,20]. Lesser-used technical indicators include energy exported to the grid [23], energy imported from the grid [23], primary energy demand [15,23] and the production-to-load ratio [14]. A variety of indicators were used for economic assessment, such as the cost of energy [12,16,17,22], net present value [14], LCOE [10,18], payback period [12] and net present cost [20,21]. Similarly, indicators of ecological impact are also varied and included reductions in *GHG* emissions [21,24], the amount of $CO_2$ emissions [10,23] and the amount of $CO_2$ equivalent emissions [22]. The wide variability in the indicators makes the interpretation and comparison of the various studies difficult.

Another observation about the indicators used in the literature is that they are homogeneously used to evaluate the goals of all actors in a community. To the best of the authors' knowledge, only one study considered that the goals and objectives can vary across actors in the energy community. Reis et al. [25] conducted a techno-economic simulation of an REC in Portugal while accounting for the different roles and goals of end users (namely, self-consuming, producing and trading and performing demand management). The study concluded that while individual actors in the energy system can have conflicting goals, self-sufficiency can be maximised, and economic benefits can incur for all stakeholders when they act collectively as an energy community. Considering energy communities as a homogeneous group with congruent goals does not reflect the diversity in motivations, goals and objectives of different actors. Therefore, studies assuming homogeneous goals for the entire community may not lead to the optimal solution for actors, and the community as a whole and a framework to evaluate these conflicting goals needs further attention.

### 1.2. Research Gap and Novelty

Energy communities and RECs have been well-researched in the scientific literature. However, as demonstrated in the state of the art, studies and models have often delved into the techno-economic aspects and, to a lesser extent, on techno-ecological and economic–ecological aspects of prosumership. Three gaps in the existing methods of assessing RECs are observed: Firstly, there is the lack of of an integrated modelling methodology to assess RECs that combines the technical, economic and ecological aspects of prosumership simultaneously. Secondly, a wide variability in the indicators used for assessing RECs makes the interpretation and comparison of the various studies difficult. Lastly, there is a lack of an assessment framework that considers the varying goals and objectives of the actors comprising an REC. Therefore, this study extends the current state of science as follows:

- We propose a single, replicable model to assess RECs across multiple disciplines.
- The model is capable of evaluating several important dimensions:
  - The technical feasibility of energy flows.
  - Quantifiable economic and ecological benefits.
  - The financial feasibility of pursuing prosumership through RECs.

- Key performance indicators (KPIs) are proposed to act as a standard set of indicators to clearly compare and interpret the results of the simulation studies.

- The KPIs also act as a framework to account for the variety in the goals and objectives of different actors of the RECs. The KPIs are mapped to the objective of each type of actor within the REC and can be used to assess whether the objective of prosumership is achieved. Aligning and conflicting objectives of the various actors can be explored, aiding decision-making processes.

The article is organised as follows: Section 2 proposes the REC Assessment Model, which is designed to evaluate the impact of the formation of an REC using three disciplines. First, in Section 2.1 the renewable energy cluster within an REC is described, followed by a description of the different actors (Section 2.2), the activities that they can pursue (Section 2.3) and the boundaries of the model (Section 2.4). This is followed by description of the three modules of the REC Assessment Model. The module on technical assessment (Section 2.5) undertakes an energy flow analysis in the decentralised energy system. It calculates the energy consumption and production in the REC based on a combination of activities allowed under the provisions of the RED II. The module on ecological assessment (Section 2.5) estimates the impact of the formation of an REC on the energy-related $GHG$ emissions of the neighbourhood. The module on economic assessment (Section 2.5) examines the economic impact of REC formation on consumers (in terms of the cost of their energy consumption) and on investors (in terms of the net present value ($NPV$) of their investment to create an REC). Lastly, in Section 3 a set of key performance indicators are defined with respect to their relevance to the actors in the REC. The discussion (Section 4) critically reflects on the methodology. Finally, a conclusion briefly summarises the main findings (Section 5) and an outlook lists possibilities for future research (Section 6).

## 2. Methodology: The REC Assessment Model

The REC Assessment model is proposed to undertake an integrated analysis of REC across three disciplines and is described in detail in the following section.

### 2.1. Decentralised Energy System

The decentralised energy system of a potential REC is used to undertake the energy flow analysis and is shown in Figure 1, along with the actors and activities possible. Emerging decentralised energy systems have the features of Renewable Energy Clusters as introduced in [26], which represent complex energy systems characterised by bi-directional energy flows, complementarity in demand or supply, involving flexibility services and including a mix of actors in terms of their scale (whether large or small) or their involvement in the energy sector (whether professional or non-professional).

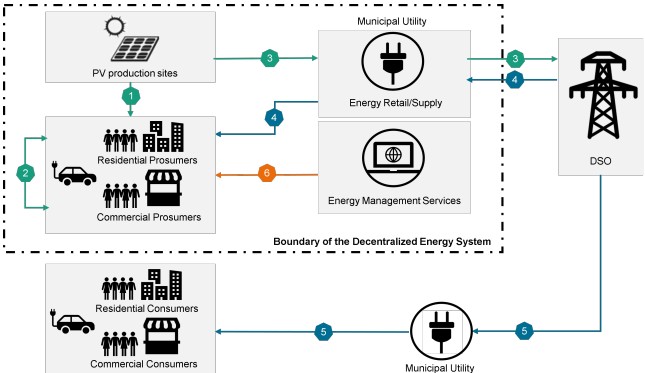

**Figure 1.** Activities and energy flows within the decentralised energy system. PV flows (in green): 1. generation and self-consumption; 2. energy sharing within the community; 3. excess feed-in to grid. Utility supply (in blue): 4. to the REC members; 5. to non-members. 6. Flexibility through demand response by shifting the timing of EV charging (in orange).

### 2.2. Actors in the REC

The REC consist of three types of actors:

- Citizens: The group of actors becomes a member of the REC, and therefore acts as prosumer and co-owner of the REC. It is possible that not all citizens participate in the REC; such citizens act as consumers.
- Investors: This group of actors invests to set up the REC but does not consume any energy generated in the REC. Therefore, they act as non-prosumer co-owners of the REC. This group of actors can consist of financial institutions (such as banks or strategic investors) or individuals (such as citizens, landlords, etc.).
- Energy System Actors: This group represents the actors who are responsible for maintaining the balance of the local energy system and managing the supply of energy to and from the REC. This may include energy suppliers (private companies or public utility companies) and the distribution system operator (DSO).

### 2.3. Activities Pursued in the REC

The possible activities that can be pursued by the actors of the REC are:

- Solar photovoltaic (PV) generation: Actors jointly invest in rooftop PV systems used to generate renewable energy.
- Self-consumption: Actors within a building consume electricity directly within the building where it is generated.
- Energy sharing: Actors share any excess energy after self-consumption to meet the demand elsewhere in the community.
- Grid feed-in: When energy generation is higher than the demand of the REC, the excess is fed into the public grid.
- Grid consumption: When PV energy is insufficient to meet the demand, the REC is supplied by procuring energy from the grid. Consumers who may not want to be a part of the REC can be supplied from the grid if they are outside the system boundary of this article.
- Flexibility through demand response: Flexibility services could be applied by the community (e.g., to maximise self-consumption by shifting loads into times of high on-site PV production). This allows the community to offer implicit flexibility (i.e., the flexibility of demand that does not interact with the energy market) [27,28].

### 2.4. System Boundaries

The analysis in the model is limited to electricity demand and supply in the REC and does not include heat demand. It has been assumed that all residents are participating in the REC and become prosumers; the consumers who are not a part of the REC and always consume grid-supplied electricity are outside the system boundary. The model analyses the energy, ecological and economic impacts at an aggregated level—for the buildings and for the REC but not for the individual residents of the REC.

### 2.5. Technical Assessment

The Technical Assessment module calculates the energy flows in the decentralised energy system and determines the sources of meeting the energy demand of the REC. Suppose the REC consists of N buildings. Each building n has a load $P_b(t, n)$ and can generate renewable power $P_{pv}(t, n)$ from its PV systems as a mean power value at timestamp $t$ within a specific measurement interval $\Delta t$. Some consumers in each building may be owners of electric vehicles (EVs) having an electric load profile $P_{ev}(t, n)$. The losses occurring from the transmission of energy from the site of generation to the site of consumption within the REC are omitted in the calculations. It is assumed that controlled charging of EVs is possible to offer flexibility through demand response, but the EVs cannot be discharged

and are, therefore, assumed to act solely as electric loads. For each building ($n = 1, 2 \dots N$) at each timestamp $t$, the total electrical load is equal to:

$$P_{load}(t, n) = P_b(t, n) + P_{ev}(t, n) \tag{1}$$

Including the building PV system, the residual load is calculated as:

$$P_{residual}(t, n) = P_{load}(t, n) - P_{pv}(t, n) \tag{2}$$

The electrical energy consumption, production and residual load for each building and timestamp follows as:

$$E_{load}(t, n) = P_{load}(t, n) * \Delta t \tag{3}$$

$$E_{pv}(t, n) = P_{pv}(t, n) * \Delta t \tag{4}$$

$$E_{residual}(t, n) = P_{residual}(t, n) * \Delta t \tag{5}$$

The time interval $\Delta t$ is the timespan between two mean power measurements and should have a reasonable resolution as a trade-off between minimising data volume and calculation time on one hand and losing relevant information on the other. High resolutions $\leq 1$ min are preferable to evaluate load peaks, while for energy balancing 15 min blocks are recommended as they go well in line with the EPEX spot market [29].

Self-consumption of energy: The electricity generated by the PV panels of the building can be consumed instantaneously in the same building behind the meter. The self-consumed energy $E_{self-cons}(t, n)$ in each building $n$ at time interval $\Delta t$ is calculated as:

$$E_{self-cons}(t, n) = min(E_{load}(t, n), E_{pv}(t, n)) \tag{6}$$

Energy sharing within the REC: If energy sharing is pursued in the REC, then buildings can share the surplus energy (after self-consumption) with other buildings which are simultaneously facing demand in excess of PV generated. Various energy dispatch strategies can be employed, such as a hierarchical/priority dispatch or a pro-quota/proportional dispatch. The hierarchical dispatch requires a priority list to be defined, and energy is shared with consumers as per their rank in the list. The pro-quota dispatch strategy shares the surplus energy with all consumers in proportion to their deficit. In the former strategy, some consumers on the lower end of the priority order may not be supplied during periods of low generation. In the latter, all consumers with a deficit are supplied by local energy, thus ensuring that all consumers fairly share the energy generated by them collectively. This model uses the pro-quota energy dispatch strategy for energy sharing and follows the methodology proposed by Moncecchi et al. [13].

The surplus $E_{surplus}$ and deficit $E_{deficit}$ of energy at time interval $\Delta t$ for each building $n$ and the whole REC are calculated as:

$$E_{surplus}(t, n) = E_{pv}(t, n) - E_{self-cons}(t, n) \tag{7a}$$

$$E_{surplus}^{rec}(t) = \sum_{n=1}^{N} E_{surplus}(t, n) \tag{7b}$$

$$E_{deficit}(t, n) = E_{load}(t, n) - E_{self-cons}(t, n) \tag{8a}$$

$$E_{deficit}^{rec}(t) = \sum_{n=1}^{N} E_{deficit}(t, n) \tag{8b}$$

The total amount of energy that can be shared within the community $E_{shared}^{rec}(t)$ at time interval $\Delta t$ is a factor of the overall energy surplus or deficit in the community. It can be calculated as the minimum of energy surplus and deficit in the community at time interval $\Delta t$.

$$E_{shared}^{rec}(t) = min(E_{surplus}^{rec}(t), E_{deficit}^{rec}(t)) \tag{9}$$

Furthermore, two energy-sharing coefficients, $k_{sup}(t)$ and $k_{col}(t)$, are used to determine the amount of energy shared within the community.

$k_{sup}(t)$: The proportion of energy requested from the community to meet the deficit which can be supplied by the energy shared within the community, defined as:

$$k_{sup}(t) = \frac{E_{shared}^{rec}(t)}{E_{deficit}^{rec}(t)} \tag{10}$$

$k_{col}(t)$: The proportion of energy offered to the community from the surplus which can be collected from the community, defined as:

$$k_{col}(t) = \frac{E_{shared}^{rec}(t)}{E_{surplus}^{rec}(t)} \tag{11}$$

In intervals when $E_{surplus}^{rec}(t)$ is greater than $E_{deficit}^{rec}(t)$, the community has a net surplus of energy, and $k_{sup}(t)$ takes the maximum value of 100%, implying that all the requested energy to meet the deficit can be supplied by the community. Similarly, in intervals when $E_{deficit}^{rec}(t)$ is greater than $E_{surplus}^{rec}(t)$, the community has a net deficit of energy, and $k_{col}(t)$ takes the maximum value of 100%, implying that all the surplus energy offered to the community is collected to meet the deficit.

Under the pro-quota dispatch strategy, the values of $k_{sup}(t)$ and $k_{col}(t)$ remain the same for each building for each time interval. For each time step, the energy shared by each building into the community $E_{shared}^{b \to rec}(t, n)$ is calculated as:

$$E_{shared}^{b \to rec}(t, n) = E_{surplus}(t, n) * k_{col}(t) \tag{12}$$

Similarly, for each time step, the amount of energy shared by the community into each building $E_{shared}^{rec \to b}(t, n)$ is calculated as:

$$E_{shared}^{rec \to b}(t, n) = E_{deficit}(t, n) * k_{sup}(t) \tag{13}$$

To ensure that each building always has an uninterrupted supply of energy, any remaining energy deficit is met through grid-supplied electricity, which is equal to:

$$E_{load}^{grid}(t, n) = E_{deficit}(t, n) - E_{shared}^{rec \to b}(t, n) \tag{14}$$

Any surplus of energy after the energy deficit in the community has been met can be fed into the grid, calculated as:

$$E_{pv}^{grid}(t, n) = E_{surplus}(t, n) - E_{shared}^{b \to rec}(t, n) \tag{15}$$

Annual energy consumption in the REC: The annual electricity consumption of the REC can be calculated by summing up the energy consumption from different sources for all buildings ($n = 1, 2 \ldots N$) and over all time steps in a year ($\Delta t = 1, 2 \ldots T$), where $T$ is the number of time intervals in a year:

$$E_{load} = \sum_{n=1}^{N} \sum_{t=1}^{T} E_{self-cons}(t, n) + E_{shared}^{rec \to b}(t, n) + E_{load}^{grid}(t, n) \tag{16}$$

The self-sufficiency share ($SSS$) of energy consumption is a metric to measure the proportion of energy demand that can be met through energy generated in the REC, expressed as:

$$SSS = \frac{\sum_{n=1}^{N} \sum_{t=1}^{T} E_{self-cons}(t, n) + E_{shared}^{rec \to b}(t, n)}{\sum_{n=1}^{N} \sum_{t=1}^{T} E_{load}(t, n)} \tag{17}$$

The PV self-consumption share (*SCS*) can be used to measure the amount of total PV generated in the REC which is consumed within the REC itself, calculated as:

$$SCS = \frac{\sum_{n=1}^{N} \sum_{t=1}^{T} E_{self-cons}(t,n) + E_{shared}^{rec \rightarrow b}(t,n)}{\sum_{n=1}^{N} \sum_{t=1}^{T} E_{pv}(t,n)} \tag{18}$$

### 2.6. Ecological Assessment

This module calculates the ecological impact of the creation of an REC by evaluating the *GHG* emissions of energy consumption. The annual *GHG* emissions of energy consumption in the REC are calculated based on the annual energy consumed in the REC and the source of generation of the energy—whether self-generated or grid-supplied.

*GHG*s are emitted at each stage of the energy generation process, from the construction of the generating unit, provision of fuel to the generator, operations to generate electricity and decommissioning of the generator at the end of its operational life. The operational emissions are directly attributed to the energy generation, while those related to upstream activities (construction and fuel provision) or downstream activities (decommissioning) can be indirectly attributed to the energy generated [30].

To evaluate the *GHG* emissions of the REC, only the direct operational emissions of energy generation are included, while the indirect emissions are omitted. The operational emissions of electricity consumed from the grid $F_{GHG}^{grid}$ are calculated based on the location-based average emission factor for the electricity grid, as per the guidelines of the Greenhouse Gas Protocol [31]. The operational emissions from grid-connected renewable energy sources (such as the PV systems under study) are negligible. As a result, the *GHG* emission factor for self-generated PV $F_{GHG}^{pv}$ can be assumed to be zero [31,32].

Based on the aforementioned assumptions and methodology, the *GHG* emissions (expressed in carbon dioxide equivalents ($CO_2$eq)) of the REC can be calculated as:

$$GHG = E_{load}^{grid} * F_{GHG}^{grid} + (E_{self-cons} + E_{shared}^{rec \rightarrow b}) * F_{GHG}^{pv} \tag{19}$$

### 2.7. Economic Assessment

The Economic Assessment module consists of studying the economic performance of an REC from two perspectives: the consumer perspective and the investor perspective. The consumer perspective quantifies the cost of energy consumption for consumers and how it is affected by pursuing prosumership and consuming lower-cost locally generated green energy. The investor perspective studies whether investing in an REC is an economically profitable decision for investors and co-owners of the REC. Note that people living within the REC can have both roles, the consumer and the investor within the REC. Therefore, each EUR saved on PV shared within the community that reduces the electricity bill will also decrease the revenue of the REC. The methodologies for both sub-modules are described below.

(a)    Consumer Perspective—The Cost of Energy Consumption:

The cost of energy consumption of the REC is calculated based on the electricity supply tariff structure. The tariff structure is assumed to consists of five components: fixed supply costs, grid usage costs, variable supply costs, taxes and retailer margin. Is is assumed that the rates of grid usage cost and taxes are the same for all types of consumers. Self-consumption of electricity is assumed to take place behind the meter and is not subject to grid usage costs.

Grid usage cost: The grid usage cost applies to the energy consumption through the public electricity grid. The grid usage cost is assumed to have two components: the performance cost $C_{gp}$ and the labour cost $C_{gl}$, which are calculated as follows:

$$C_{gp} = max(P_{residual}(t)) * F_{gp} \tag{20}$$

$$C_{gl} = (E_{load}^{grif} + E_{shared^{rec \rightarrow b}}) * F_{gl} \tag{21}$$

with $F_{gp}$ being the costs for providing annual peak power in EUR/kW and $F_{gl}$ being the costs for providing energy in EUR/kWh. The total annual grid usage costs are the sum of labour and performance cost:

$$C_g = C_{gl} + C_{gp} \tag{22}$$

Fixed supply cost: The fixed supply cost $C_{sf}$ is applied by the retailer to keep the consumers connected and supplied with electricity as well as for measuring point operation. It is a fixed amount that is paid monthly or annually by each household, commercial unit or workspace. The fixed supply cost is calculated by multiplying the number of rental units $n_{units}$ in the building with the fixed supply factor $F_{sf}$

$$C_{sf} = n_{units} * F_{sf} \tag{23}$$

Variable supply cost: The variable supply cost $C_{sv}$ is charged by the retailer for the actual electricity consumed. The variable supply tariff is calculated by multiplying the energy consumed from the public grid with the variable supply factor $F_{sv}$.

$$C_{sv} = E_{load}^{grid} * F_{sv} \tag{24}$$

Taxes: The energy taxes are levied on the amount of electricity consumed and are calculated by multiplying with the tax rate $F_{tax}$:

$$C_{tax} = E_{load} * F_{tax} \tag{25}$$

The formation of RECs can be incentivised by exemptions on components of variable taxes, leading to a special tax rate $F_{tax}^{rec}$. As a result, the cost of variable tax for RECs can be calculated as:

$$C_{tax} = E_{load}^{grid} * F_{tax}^{rec} \tag{26}$$

Retailer margin: The retailer margin is the profit that the retailer earns for offering the service of supplying electricity to consumers. The retailer margin has two components: variable retailer margin $C_{rmv}$, which varies with the electricity consumption, and the fixed retailer margin $C_{rmf}$, which is a fixed value for the consumer. The retailer margin is given by the following equations:

$$C_{rmv} = E_{load} * F_{rmv} \tag{27}$$

$$C_{rmf} = n_{units} * F_{rmf} \tag{28}$$

$$C_{rm} = C_{rmf} + C_{rmv} \tag{29}$$

Total annual cost of energy consumption: Finally, the total annual cost of energy consumption is calculated as a sum of its components:

$$C = C_g + C_{sf} + C_{sv} + C_{tax} + C_{rm} \tag{30}$$

(b)  Investor Perspective—*NPV* of Investing in an REC:

An economic feasibility analysis is conducted to determine if forming an REC is an acceptable investment decision using the well-established net present value (*NPV*) rule. The *NPV* rule governs that a project with a positive *NPV* should be accepted, while one with a negative *NPV* should be rejected.

The *NPV* analysis aims to test if the future earnings of the REC from various sources of revenues and incentives can cover the cost of the initial investment. The investment for establishing the REC comes from equity and loan, with loan repayment commencing at the end of year 1 of the analysis. The capital expenditure (CAPEX) is assumed to include the costs of PV modules, inverters and balance of system costs, which include costs of installation, mounting, DC cabling, switchgear, grid connection, transformer, infrastructure planning and documentation [33,34]. It is assumed that the electricity is sold in the wholesale electricity market once the incentive period is over. The annual revenues,

costs, incentives and wholesale electricity prices are assumed to remain constant over the period of analysis unless stated otherwise.

The net present value of the project is given as:

$$NPV = \sum_{y=1}^{T} \frac{CF(y)}{(1+d)^y} \qquad (31)$$

where $CF(y)$ is the cash flow in year $y$ adjusted for inflation, $d$ is the discount rate or the rate of return that can be earned in alternate investments and $T$ is the expected lifetime of the project. The cash flow for year $y$ over the project lifetime is the difference of net income $I_{net}$ and the loan repaid *Loan*:

$$CF(y) = I_{net}(y) - Loan(y) \qquad (32)$$

where

$$I_{net}(y) = Revenue(y) + Inc(y) - OPEX_{rec}(y) - Interest(y) - Tax(y) \qquad (33)$$

The terms used to calculate $CF(y)$ are explained below:

*Revenue(y)*: Based on whether the cost savings of prosumership are directly and fully passed on the prosumers or not, the revenue accrued to REC will be different. When all costs savings are passed to the prosumers, the revenue of the REC is the same as $C_{rm}$ calculated in Equation (29). However, when cost savings are not passed on to the prosumers they are retained with the REC and, with these increased earnings, can be used to pay back the investment loan faster. In this case, the total cost of energy to consumers remains constant, as per the business-as-usual, and any reduction in cost adds to the retailer margin of the REC.

*Inc(y)*: REC earns financial incentives, e.g., in the form of feed-in-tariff for electricity supplied to the grid or other incentives offered by the Member State.

$$Inc_{feedin}(n, y) = E_{pv}^{grid}(n, y) * F_{feedin} \qquad (34)$$

$$Inc(n, y) = Inc_{feedin}(n, y) + \ldots \qquad (35)$$

*Interest(y)*: The interest to be paid on the outstanding loan amount subject to interest rate $r$.

$$Interest(y) = r * Loan_{outstanding}(y) \qquad (36)$$

*Tax(y)*: REC pays the corporate income tax on its taxable income. Tax is not paid if the taxable income is negative.

$$Tax(y) = max(0, taxrate(y) * (Revenue(y) + Inc(y) - OPEX_{rec}(y) - Interest(y))) \qquad (37)$$

## 3. Key Performance Indicators for RECs

The results of simulating an REC using the proposed REC Assessment Model can be interpreted in several ways. To streamline the interpretation and comparison of results, a set of key performance indicators (KPIs) are defined. Since each actor within the REC has different objectives that motivate them to participate in an REC, the KPIs that are relevant for one group of actors may not hold relevance for others. Therefore, the KPIs that are most relevant to each group of actors are identified.

The following KPIs are proposed to assess the REC and evaluate its benefits for various actors (shown in Figure 2):

- For the Citizens, who are prosumers and co-owners of the REC, the most important KPIs are the self-sufficiency share ($SSS$) (Equation (18)), the cost of energy (Equation (30)) and the $GHG$ emissions avoided (Equation (19)). For this group of actors, the best setup of the REC is the one that offers the highest $SSS$, the lowest cost of energy and the greatest $GHG$ emissions avoided.

- For the Investors who fund the setting up of the REC, i.e., the non-prosumer co-owners of the REC, the $NPV$ (Equation (31)) of the project is the most important indicator to evaluate the investment opportunity. The best set up of the REC for investors will be the one with the highest positive $NPV$.
- For the Energy System Actors, namely the energy supplier and the DSO, both parties aim to achieve maximum self-consumption share ($SCS$) (Equation (17)) of the locally generated energy, as this indicates higher consumption of local energy, lower losses in transmission and distribution and more efficient use of the energy system. For the energy supplier, the highest positive $NPV$ (Equation (31)) is another important indicator suggesting the profitability of the investment. For the DSO, grid capacity constraint is an important consideration that can be assessed by the lowest peak load in the network (Equation (2)).

| Actors within the REC | | Key Performance Indicator |
| --- | --- | --- |
| | Citizens (prosumer co-owners) | Highest self-sufficiency share |
| | | Lowest annual cost of energy |
| | | Lowest annual GHG emissions |
| | Investors (non-prosumer co-owners) | Highest positive NPV |
| | Energy System Actors | Highest self-consumption share |
| | | Highest positive NPV |
| | | Lowest peak load |

**Figure 2.** KPIs relevant to various actors within the REC.

These KPIs can be used to compare different combinations of prosumership activities undertaken in the REC. The KPIs can enable different actors within the REC to identify the system setup that meets their objective and offers them the highest benefit. The KPIs can identify areas where the objectives of the actors are aligned and where they are in conflict. These KPIs can be used to compare different RECs planned under the same regulatory framework to identify the one offering the highest techno-economic, socioeconomic, ecological or other benefits per unit of investment.

## 4. Discussion

The aim of this study is to propose an integrated model that combines technical, economic and ecological analysis to assess the various possibilities of prosumership within an REC and to offer a framework to compare and interpret the results of the simulation. The proposed model and KPIs are not without limitations and offer several areas for future research. The model assumes that all members of the community join the REC, however, this may not always be the case; this functionality can be added to the energy flow analysis in future research. The model also assumes that all energy sharing takes place through the public grid. In cases where energy sharing cannot be facilitated through the public grid, a private grid for the REC can be considered; in such a case, the capital costs of the private grid should be included in the cash flow calculations (Equation (32)).

With regard to the technical analysis, the model assumes that data are consistently available with a reasonable resolution (of at least 15 min). However, gathering this data is not addressed within this section. If not present already, installing smart metering infrastructure can significantly increase investment and operational costs. In [35], the author found that in the United States, the willingness to install smart meters decreases by 18% with each USD 2.50 spent monthly. So, the costs of installing smart meter infrastructure should be as small as possible (e.g., by only having one smart meter at the grid connection point of a house and not on a household level), while the benefits of the measurement infrastructure need to be emphasised for REC members.

The initial investment for establishing the REC is assumed to come from a mix of equity (contributed by prosumer co-owners) and loan (from non-prosumer co-owners). While *NPV* has been proposed as a KPI of financial feasibility for non-prosumer co-owners, a suitable financial KPI for prosumer co-owners is lacking in the current article. Furthermore, the inclusion of *NPV* as a KPI can be questioned since the RED II intends RECs to provide environmental, economic or social community benefits for their members or for the local areas where they operate, rather than financial profits. However, *NPV* has been included as it is widely used by investors to evaluate investment opportunities and helps them in comparing RECs with other projects in terms of the return on investment.

The adapted model should undergo validation such as through expert interviews, structured walkthroughs or other methods proposed in [36].

## 5. Conclusions

Collective prosumership by citizens participating in Renewable Energy Communities (RECs) is envisioned to be an essential pillar of the clean energy transition in the EU. Developing a holistic understanding of the possibilities and benefits of collective prosumership through RECs is essential to foster their development and drive their uptake. In this paper, we propose a methodology to assess RECs across multiple disciplines. By combining technical, economic and ecological analysis, we offer a single and replicable model that can be used to assess the possibilities and benefits of RECs. The model consists of three modules: First, the technical possibilities of collective prosumership in a potential REC are calculated using energy flow analysis. Second, the economic impact of prosumership is evaluated from the perspective of two actors within the REC: for consumers, this is expressed in terms of the annual cost of energy consumption, and for investors, in terms of the net present value of their investment in the REC. The third module assesses the impact of prosumership on annual greenhouse gas emissions of energy consumption, expressed in $CO_2$ equivalent. A set of key performance indicators (KPIs) that are relevant to the objectives of different actors within the REC are mapped to the equations within the assessment framework. The KPIs can be used to compare the impact of combinations of various prosumership activities within the same REC or to compare two different RECs on the benefits offered (technical, economic or ecological) vis-a-vis the investments incurred. The KPIs also offer insights into the aligning and conflicting objectives of the stakeholders of the REC. While energy communities and other citizen-led energy initiatives have been well-researched, the proposed method could help in developing a more holistic, integrated and multi-disciplinary knowledge of RECs as defined under the latest EU Directives.

## 6. Outlook

Adapting and replicating the model in member states where the RED II has fully or partially been transposed (such as in Belgium (Flanders), Italy, France, Austria, Ireland, Denmark, Lithuania, Spain and Portugal [8]) could make the approach more widely applicable across the EU. An adaptation of the presented general assessment framework within the rather restricted German regulatory environment is planned to be conducted within a consecutive publication. Such regulatory analyses together with organisational aspects mark important additions to the technical, economic and ecological implications of forming an REC and should therefore be included in the feasibility analysis within future work. An extension of this model to include flexibility within the technical assessment could be added in future work. Several case studies on energy communities in this context deal with individual or shared energy storages and highlight their added value in a community context [17,19,20]. A methodology to quantify the flexibility of different technologies (electric vehicles, batteries, heat pumps and combined heat and power units) as storage equivalents that can be aggregated within a neighbourhood was conducted in [24,37]. Following such an approach, the potential of flexibility could be quantified with regard to power, energy and time and added to the proposed model for application in specific energy system services. Lastly, since the number of members and level of participation of citizens

in a district/neighbourhood while forming an REC can have implications on the energy flows, economic assessment and *GHG* emissions, this should be a line of inquiry pursued in future research.

**Author Contributions:** Conceptualisation, S.C. and A.S.; data curation, A.S.; formal analysis, M.K. and F.P.; funding acquisition, A.S. and M.K.; investigation, S.C. and A.S.; methodology, S.C. and A.S.; project administration, A.S.; resources, A.S.; software, A.S.; supervision, A.S.; validation, S.C. and A.S.; visualisation, S.C. and A.S.; writing —original draft, S.C. and A.S.; writing—review and editing, M.K. and F.P. All authors have read and agreed to the published version of the manuscript.

**Funding:** The work presented has been conducted in the EnStadt:Pfaff project. The research leading to these results has received funding from the German Ministry of Education and Research (BMBF) under the funding number 03SBE112G and the Ministry for Economics and Energy (BMWi) under the funding number 03SBE112D.

**Data Availability Statement:** Data will be made available upon request.

**Acknowledgments:** The research scope of this study was developed within the Fraunhofer Cluster of Excellence Integrated Energy Systems (CINES).

**Conflicts of Interest:** The authors declare no conflict of interest. The funders had no role in the design of the study; in the collection, analyses, or interpretation of data; in the writing of the manuscript; or in the decision to publish the results.

## Abbreviations

The following abbreviations are used in this manuscript:

| | |
|---|---|
| $CO_2$ | Carbon Dioxide |
| $CO_2$ eq | Carbon Dioxide Equivalent |
| CEC | Citizen Energy Community |
| CF | Cashflow |
| DSO | Distribution System Operator |
| EC | Energy Communities |
| EU | European Union |
| *GHG* | Greenhouse Gas |
| KPI | Key Performance Indicator |
| LCOE | Levelised cost of electricity |
| *NPV* | Net Present Value |
| PV | Photovoltaic |
| RE | Renewable Energy |
| REC | Renewable Energy Community |
| RED | Renewable Energy Directive |
| RES | Renewable Energy Sources |
| *SCS* | Self-Consumption Share |
| *SSS* | Self-Sufficiency Share |

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
