# Peer review of "Renewable Energy Communities as Modes of Collective Prosumership: A Multi-Disciplinary Assessment, Part I—Methodology"

_energies, doi:10.3390/en15238902_

Round 1
Reviewer 1 Report
The article is interesting and deals with very important issues. The following comments are expressed with kindness in order to obtain even better effect and interest of the readers.
Please extend the introduction and add the characteristics of the market in Germany for which this model is designed. In the introduction, please describe the following issues:
Are there any mechanisms supporting energy communities in Germany? (billing preferences, discounts, lower charges for energy distribution, etc.)
How many energy communities are there, what is their nature and how do they function? (energy cooperatives, or maybe larger clusters, virtual prosumer, etc.)
On pages 47-70. The authors cite positive effects of community implementation in Austria, Italy and Portugal. Perhaps it is also worth citing the results of research for the countries neighboring with Germany, e.g. https://doi.org/10.3390/en14020319; https://doi.org/10.3390/en14217113
Please extend the description of figure 2 + line 340 with information on the benefits and risks of the functioning of the energy community for the DisCo.
Have the authors made calculations and simulations on the basis of the formulas presented in the article? If so, please put the results in the open access formula.
Are the authors leading a role for energy storage in the energy community? - it is worth referring to in the final conclusions
Author Response
Please see the attachment. Note that all line numbers refer to the revised document.

Reviewer 2 Report
To start with, I would like to thank authors for their work in terms of interesting topic, high quality research and well written article(s) and want to mentioned the research conducted by authors is described in two articles energies-2049923 and energies-2049926.
The research is devoted to the development of integrated model to assess potential Renewable Energy Communities in the Member States of the European Union through analyzing the energy, ecological and economic impacts for the buildings and for Renewable Energy Communities.
Overall, the research is at high level and I have no any concerns to it at all (small typos will be eliminated by language editors). However, I see one principal problem. The research is as voluminous enough so authors decided to divide it into two articles the current one (energies-2049923) and energies-2049926. The decision caused the following problem:
The current paper (energies-2049923) contains only the theoretical results of the Model building without the example of model application \ the comparison with others through the case study. While the second paper (energies-2049926) provides the case study of the model application in Germany on the theoretical research presented in the first one. So, without each other they can not be considered as of full value. However, readers of one paper can be not aware of the existing of other one and miss it. I cannot see the solution to cite papers in each other, except for the publication of the second paper after the first one. But, anyway, the problem will be half ameliorated.
As papers describe one research and they must have the accepted article structure, some information is doubled and can be considered as plagiarism. Please, remember your statement in cover letter – “We confirm that neither the manuscript nor any parts of its content are currently under consideration or published”
I recommend authors to redesign the whole research in one paper or solve the problem with editors.
The same review will be for the both papers (energies-2049923 and energies-2049926.)
Author Response

(The authors gave the same response as above.)

Reviewer 3 Report
The authors propose an integrated model to undertake a multi-disciplinary assessment of a potential renewable energy community. The model analyses the technical possibilities of collective prosumership based on consumption and generation profiles, it evaluates the economic impacts of prosumership and quantifies the ecological impact of prosumership.
The topic is very relevant, methodology, modules and indicators well presented. Overall, the paper is well written and easy to read.
- Given the development of REN, the goal to implement a methodology to assess RECs across multiple disciplines should be more motivated by highlighting the contributions of the paper: please, highlight the novelty of the paper by providing a point-by-point contribution at the end of the introduction section;
- The “Introduction section” has to be updated and strengthened. In order to understand the state-of-the-art in this field in a better way, the most relevant background information should be provided. Especially, in order to give the readers a sense of continuity, a literature check of the relevant and recent papers published on the topics dealt with the manuscript in the recent year is needed, and the content of relevant papers should be related to the results and findings presented in your publication. The Authors should also consider and discuss in the introduction section the following papers and the items addressed:
“Edge Computing Parallel Approach for Efficient Energy Sharing in a Prosumer Community” L Scarcello, A Giordano, C Mastroianni - Energies, 2022
Prosumer communities and relationships in smart grids: A literature review, evolution and future directions E Espe, V Potdar, E Chang - Energies, 2018
- In section 2.1.5. Technical evaluation, Δt is introduced to quantify the amount of energy exchanged, starting from the power values. how to quantify this parameter? This parameter must be opportunity set in order to make the power values ​​comparable. Have you carried out a study to estimate a maximum value of Δt below which this calculation does not lose validity?
- Conclusion section should be extended using (i) the main results obtained in the paper, (ii) the limitation of the proposed model, and (iii) perspective of the future research. Is the implementation of the model planned on a real test case?
Author Response
Please see the attachment. Please note that line numbers refer to the revised document.

Round 2
Reviewer 1 Report
Thank you for considering the reviewer's suggestions.
The article is very interesting and ready for publication.
Reviewer 2 Report
Dear authors,
Thank you for their huge work spent on the revision and addressing comments of mine and other reviewers!
I have already told that I liked the paper in terms of the quality research and the paper design. Concerning to merging both papers, I fully agree that purposes of each paper are different and might be separated. I am really glad that you can have negotiated this question with editorial office and you have added additional information on mentioning each paper.
I recommend your paper for publication and wish you continue working on such qualitive research